# Information-based Adaptive Stimulus Selection to Optimize Communication Efficiency in Brain-Computer Interfaces

**Boyla O. Mainsah,**[1] **Dmitry Kalika,**[2] **Leslie M. Collins,**[1,*]
**Siyuan Liu,**[1] **Chandra S. Throckmorton**[1]
[1]Department of Electrical and Computer Engineering, Duke University, Durham, NC, USA
[2]Johns Hopkins University Applied Physics Laboratory, Laurel, MD, USA
[*]Corresponding author: `leslie.collins@duke.edu`

## Abstract

Stimulus-driven brain-computer interfaces (BCIs), such as the P300 speller, rely on using a sequence of sensory stimuli to elicit specific neural responses as control signals, while a user attends to relevant target stimuli that occur within the sequence. In current BCIs, the stimulus presentation schedule is typically generated in a pseudo-random fashion. Given the non-stationarity of brain electrical signals, a better strategy could be to adapt the stimulus presentation schedule in real-time by selecting the optimal stimuli that will maximize the signal-to-noise ratios of the elicited neural responses and provide the most information about the user's intent based on the uncertainties of the data being measured. However, the high-dimensional stimulus space limits the development of algorithms with tractable solutions for optimized stimulus selection to allow for real-time decision-making within the stringent time requirements of BCI processing. We derive a simple analytical solution of an information-based objective function for BCI stimulus selection by transforming the high-dimensional stimulus space into a one-dimensional space that parameterizes the objective function - the prior probability mass of the stimulus under consideration, irrespective of its contents. We demonstrate the utility of our adaptive stimulus selection algorithm in improving BCI performance with results from simulation and real-time human experiments.

## 1   Introduction

Brain-computer interfaces (BCIs) acquire brain signals in real-time, process the signals to extract relevant neural information and translate this information into commands that convey a user's state or intent to control external devices [1]. BCIs are typically defined by the specific neural signal components that are used to control the BCI [2]. The generation of these neural signals can either be self-initiated, such as with motor imagery, or elicited with sensory stimuli, such as is the case with BCIs based on *event-related potentials* (ERPs). BCIs can replace or restore control abilities in individuals with severe motor disabilities such as amyotrophic lateral sclerosis (ALS) or spinal cord injury [3]. The P300 ERP-based BCI speller [4] is one of the most widely researched BCIs for communication for individuals whose severe neuromuscular limitations preclude their use of most commercially available assistive technologies, such as individuals with late stage ALS [1, 5]. The P300 speller relies on eliciting and detecting ERP responses embedded in electroencephalography (EEG) data while a user attends to relevant target visual, tactile or auditory stimuli occurring within a sequence of non-relevant (background) stimuli. In a typical visual P300 speller, a user focuses on a desired character displayed on an interface with $M$ possible options, such as in Figure 1(a), while groups of characters, termed *flash groups*, are sequentially illuminated on the screen. Ideally,

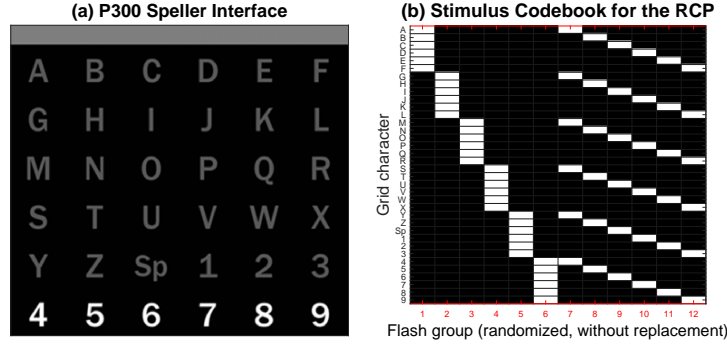

Figure 1: (a) Example P300 speller interface. (b) Stimulus codebook associated with the corresponding row-column paradigm (RCP). Each column in the codebook represents a flash group with the characters highlighted in white. For example, the illuminated row flash group in (a) is flash group 6 in (b). Each row represents the presentation pattern or *codeword* of a character.

a P300 ERP is elicited when the target character is presented or "flashed." After multiple stimulus presentations, the BCI analyzes the user's EEG data with an automated algorithm to make a decision about the user's intended character.

Unfortunately, current stimulus-driven BCIs are limited by their slow communication rates due to the inherent limitations of relying on sensory stimulation and noisy EEG data to control the BCI. Multiple data measurements for each character being spelled are needed to increase the signal-to-noise ratios (SNRs) of the ERPs to facilitate their detection accuracy in noisy EEG data. Also, ERPs exhibit high temporal variability and are susceptible to psychophysical factors, such as refractory effects, user distractions and fatigue, each of which negatively impact the ERP elicitation process [6, 7]. The ability to modulate ERP responses via the stimulus presentation schedule [8] provides a mechanism by which to improve BCI performance. However, conventional BCI stimulus paradigms rely on pseudo-randomly generated stimulus presentation schedules, where the design process is often based on simplicity. A simple way to design a stimulus presentation paradigm is to group characters by rows and columns of a grid and present them in random order, which is commonly known as the *row-column* (RC) paradigm [4] (see Figure 1(b)). However, the RC paradigm is susceptible to psychophysical factors that negatively impact BCI performance. Refractory effects, for example, occur when there is a short time interval between the target character's presentation, or the *target-to-target interval* (TTI), which reduces the ERP SNR [7]. Due to random stimulus selection, a significant proportion of characters in the RC paradigm have short TTIs, and this exacerbates refractory effects. Also, users report more visual discomfort with the RC paradigm when characters adjacent to the target character are flashed together with the target [9].

Other stimulus presentation paradigms have been developed to address the limitations of the RC paradigm by using heuristics with the goal of reducing known sources of errors during ERP elicitation. The checkerboard paradigm [9] uses a set of heuristics on characters in a grid interface to prevent adjacent characters from flashing together and impose a minimum TTI between character presentations. Other research groups have investigated optimizing the stimulus presentation schedule in advance using error-correcting codes, where the Hamming distance between character stimulus presentation patterns, or *codewords*, are maximized to increase error recovery during the decoding process. However, maximizing Hamming distances favors the selection of character presentation patterns with short TTIs, which increases refractory effects. Hence, the use of error-correcting codes is unlikely to improve BCI performance [10, 11, 12], unless the impact of refractory effects are factored into the codebook design process [13]. Overall, all of these previously considered stimulus paradigms do not necessarily provide the best approach to maximize BCI performance as not every stimulus provides the same amount of utility to estimate the user's intent based on the currently observed data. Despite its limitations, the RC paradigm, which was used in the original P300 speller study [4], still remains the standard stimulus paradigm that is used in the BCI literature [14].

Alternatively, the BCI stimulus selection process can be adapted in real-time based on the data that is currently measured to maximize the elicited neural responses [15, 16]. Adaptive stimulus selection is achieved by solving a combinatorial optimization problem with an objective function that quantifies the usefulness of a stimulus-response pair to estimate the user's target character given the

uncertainties in the data. The stimulus selection process is subject to real-time BCI system constraints as the selection process from a space of $2^M$ possible flash groups needs to be executed within the time interval between two consecutive stimuli (known as the inter-stimulus interval, ISI), which typically ranges from 60-250ms in most BCI systems. A previous P300 speller study that utilized a partially observed Markov decision process for adaptive stimulus selection restricted the search space to just row or column flash groups to obtain a more tractable solution [17]. A better strategy is to allow the stimulus selection process the flexibility to explore the whole stimulus space to dynamically create flash groups to maximize the objective function, which requires a computationally efficient algorithm.

We hypothesize that the limited development of adaptive stimulus selection algorithms in the current BCI literature [18] is likely due to the lack of objective functions with tractable solutions (the "curse of dimensionality") to allow for real-time algorithm implementation. We have developed a simple, yet powerful analytical solution to an objective function, which allows for computational efficiency in exploring the high dimensional BCI stimulus space: the objective function is parameterized by the prior probability mass of a future stimulus under consideration, irrespective of its content. The main contributions of this work are as follows: 1) We introduce an adaptive BCI stimulus selection algorithm, where the objective is to maximize the information content elicited from future stimuli in estimating the user's intent, based on the neural responses being measured and the BCI's current belief regarding the user's intent; 2) We outline considerations and practical steps for real-time implementation of our adaptive BCI stimulus selection algorithm; and 3) We present preliminary results from simulation and human BCI experiments that demonstrate the potential to obtain significant performance improvements with our adaptive stimulus selection algorithm.

## 2 Adaptive Stimulus Selection Algorithm

### 2.1 P300 BCI Speller Overview

A user can select one of $M$ choices using the P300 speller. To make a selection, the user focuses on a desired target character, $C^*$, on a computer screen (such as the one shown in Figure 1(a)) while subsets of characters or flash groups are sequentially illuminated. Let $\boldsymbol{f}_t = [f_{t,1}, ..., f_{t,M}]$ represent a binary $M$-dimensional vector where each bit indicates whether a character is ($f_{t,m} = 1$) or is not ($f_{t,m} = 0$) present within a flash group. The user's EEG response to a flash group presentation is processed and scored with a user-specific classifier, yielding a score, $y_t$. The classifier score is used to update a decoding function that quantifies the possibility of each of the BCI choices to be the user's target character given the observed data. A stopping rule is used to determine when to terminate data collection; the standard approach is to collect a fixed amount of data, termed static stopping. After data collection, the character with the maximum score is selected as the target character estimate, $\hat{C}^*$.

For the decoding function, we use the naive Bayesian dynamic stopping algorithm developed in [19] to quantify the probability of each possible character being the target character after each flash group presentation. Dynamic stopping (DS) is achieved by setting a probability threshold, $P_{th}$, to terminate data collection; hence the quality of data is used to balance the trade-off between minimizing the amount of data collection to increase spelling speed and maximizing the amount of data collection to increase accuracy. The character probabilities at time index $t$ are updated accordingly:

$$P(C^* = m | \boldsymbol{y}_t, \mathbf{f}_t) = \frac{p(y_t | C^* = m, \mathbf{f}_t) P(C^* = m | \boldsymbol{y}_{t-1}, \mathbf{f}_{t-1})}{\sum_j p(y_t | C^* = j, \mathbf{f}_t) P(C^* = j | \boldsymbol{y}_{t-1}, \mathbf{f}_{t-1})}, \tag{1}$$

$$p(y_t | C^* = m, \mathbf{f}_t) \triangleq l_{t,m}(y_t) = \left\{ \begin{array}{l} l0(y_t), \text{ if } f_{t,m} = 0 \\ l1(y_t), \text{ if } f_{t,m} = 1 \end{array} \right. , \tag{2}$$

where $P(C^* = m | \boldsymbol{y}_t, \mathbf{f}_t)$ is the posterior probability that character $m$ is the target character, $C^*$, given the presented flash groups, $\mathbf{f}_t = [\boldsymbol{f}_1, ..., \boldsymbol{f}_t]$, and classifier scores, $\boldsymbol{y}_t = [y_1, ..., y_t]$; $P(C^* = m | \boldsymbol{y}_{t-1}, \mathbf{f}_{t-1})$ is the prior probability; $p(y_t | C^* = m, \boldsymbol{f}_t)$ is the likelihood of generating the classifier score, $y_t$, given character $m$ is the target character and the current flash group sequence, $\boldsymbol{f}_t$; $l0$ and $l1$ are the class conditional classifier score probability density functions (pdfs) for non-target and target stimulus events, which are estimated for each BCI user during system calibration. The denominator term in (1) represents the marginal probability of the classifier score conditioned on the current data:

$$p(y_t | \boldsymbol{y}_{t-1}, \mathbf{f}_t) = \sum_{j=1}^{M} p(y_t | C^* = j, \mathbf{f}_t) P(C^* = j | \boldsymbol{y}_{t-1}, \mathbf{f}_{t-1}). \tag{3}$$

This alternative expression for the marginal probability, $p(y_t|\boldsymbol{y}_{t-1}, \mathbf{f}_t)$, will be useful in simplifying the objection function that is used in our adaptive stimulus selection algorithm.

## 2.2 Objective Function

The information encoded in the user's EEG response to a stimulus is condensed into a classifier score, which is used to update the BCI system's current belief regarding the user's target character. Due to noise, multiple character-specific presentation patterns can generate the same sequence of classifier scores, which can lead to BCI decoding errors. The more information about the target character that is contained in the classifier scores, the more likely it is that the target character will be correctly identified. To facilitate correct target character estimation in as few stimulus presentations as possible, we use an information-based criterion [20] to bias the stimulus selection process towards stimuli that provide the most information to the BCI to correctly estimate the user's intent given the current data.

Mutual information is a non-negative measure in information theory that quantifies the amount of information about a random variable that can be obtained from another random variable [22]. It is typically denoted as $I(A; B)$, where $A$ and $B$ represent two random variables. The utility of a hypothetical future flash group, $\boldsymbol{f}_{t+1}^h$, can be estimated by evaluating the mutual information between the target character, $C^*$, and a hypothetical classifier score, $Y_{t+1}^h$, that is generated in response to the presentation of $\boldsymbol{f}_{t+1}^h$. This amount of information can also be evaluated within the context of how much the currently observed data (i.e., the previously observed flash groups, $\mathbf{f}_t$, and classifier scores, $\boldsymbol{y}_t$) reduce the uncertainty about the target character estimate. If $A$ and $B$ represent continuous and discrete random variables, respectively, and $r$ represents a specific value of another random variable, $R$, the mutual information between $A$ and $B$ conditioned on $r$ is calculated accordingly:

$$I(A; B|r) = \int_{-\infty}^{\infty} p(a|r) \left[ \sum_b P(b|a, r) \log\left(\frac{P(b|a, r)}{P(b|r)}\right) \right] da|r. \tag{4}$$

Assuming the Bayesian algorithm (1)-(3), we use the following objective function based on mutual information to select a flash group at each time step accordingly:

$$
\begin{aligned}
I(Y_{t+1}^h; C^*|\boldsymbol{y}_t, \mathbf{f}_{t+1}^h) &= \int_{-\infty}^{\infty} p(y_{t+1}^h|\boldsymbol{y}_t, \mathbf{f}_{t+1}^h) \left[ \sum_{m=1}^{M} P^h(C^* = m|\boldsymbol{y}_{t+1}^h, \mathbf{f}_{t+1}) \times \right. \\
&\quad \left. \log\left(\frac{P^h(C^* = m|\boldsymbol{y}_{t+1}^h, \mathbf{f}_{t+1})}{P(C^* = m|\boldsymbol{y}_t, \mathbf{f}_t)}\right) \right] dy_{t+1}^h|\boldsymbol{y}_t, \mathbf{f}_{t+1}^h \\
&= \int_{-\infty}^{\infty} \left[ \sum_{m=1}^{M} l_{t+1,m}(z_{t+1}^h) p_{t,m} \log\left(\frac{l_{t+1,m}(z_{t+1}^h)}{\sum_{c=1}^{M} l_{t+1,c}(z_{t+1}^h) p_{t,c}}\right) \right] dz_{t+1}^h, \tag{5}
\end{aligned}
$$

$$\boldsymbol{f}_{t+1}^s = \underset{\boldsymbol{f}_{t+1}^h}{\operatorname{argmax}} I(Y_{t+1}^h; C^*|\boldsymbol{y}_t^h, \mathbf{f}_{t+1}^h), \; \forall \boldsymbol{f}_{t+1}^h \in \mathcal{F}, \tag{6}$$

where $I(Y_{t+1}^h; C^*|\boldsymbol{y}_{t+1}^h, \mathbf{f}_{t+1}^h)$ is the mutual information between the future classifier score $Y_{t+1}^h$ and the target character $C^*$ conditioned on the current classifier scores, $\boldsymbol{y}_t$, and hypothetical future flash group sequence, $\mathbf{f}_{t+1}^h = [\mathbf{f}_t, \boldsymbol{f}_{t+1}^h]$; $\mathcal{F}$ is the search space of all possible flash groups; $z_{t+1}^h \triangleq y_{t+1}^h|\boldsymbol{y}_t, \mathbf{f}_{t+1}^h$ for notational simplicity; and $\boldsymbol{f}_{t+1}^s$ is the flash group that maximizes the objective function, which is selected for presentation. Alternatively, the mutual information can be expressed as an expectation of the Kullback-Leibler divergence $(D_{KL})$ between the hypothetical posterior $(p_{t+1}^h)$ and current $(p_t)$ character probability distributions taken over the future conditional classifier score, $z_{t+1}^h$ [20, 21],

$$I(Y_{t+1}^h; C^*|\boldsymbol{y}_t, \mathbf{f}_{t+1}^h) \triangleq E_{z_{t+1}^h}\left[D_{KL}(p_{t+1}^h||p_t)\right]. \tag{7}$$

The conditional mutual information in (5) is not the most convenient formulation to solve the optimization problem due to the lack of a closed form solution. When using gradient-based methods in a high dimensional space (i.e., in terms of the $M$-dimensional space, $\mathcal{F}$) where the solution to the objective function is not tractable, convergence within a time limit (which is vital for BCI real-time

operation) is not guaranteed and there is the possibility to be stuck in a local maximum. Alternatively, we can exploit the binary choice in the character likelihood assignments (2) to group together flashed and non-flashed characters in the flash group under consideration, $\boldsymbol{f}_{t+1}^h$. With this grouping, the denominator of the logarithm term in (5) can be expressed as:

$$\sum_{c=1}^{M} l_{t+1,c}(z_{t+1}^h)p_{t,c} = \left[ l0(z_{t+1}^h) \sum_{\forall c:f_{t,c}=0} p_{t,c} \right] + \left[ l1(z_{t+1}^h) \sum_{\forall c:f_{t,c}=1} p_{t,c} \right]$$
$$= l0(z_{t+1}^h)(1 - P1_t(\boldsymbol{f}_{t+1}^h)) + l1(z_{t+1}^h)P1_t(\boldsymbol{f}_{t+1}^h) \tag{8}$$

$$P1_t(\boldsymbol{f}_{t+1}^h) = \sum_{\forall c:\boldsymbol{f}_{t+1,c}^h=1} p_{t,c} \tag{9}$$

where $P1_t(\boldsymbol{f}_{t+1}^h)$ is the *sum of prior probabilities* at time $t$ for characters that are flashed in $\boldsymbol{f}_{t+1}^h$, which we will denote as $P1_t$ for simplicity. Similarly, we group together flashed and non-flashed characters in the discrete integral in (5) to obtain the following expression for the mutual information:

$$I(Y_{t+1}^h; C^*|\boldsymbol{y}_t, \mathbf{f}_{t+1}^h) = \int_{-\infty}^{\infty} \mathcal{I}(z_{t+1}^h)dz_{t+1}^h \tag{10}$$

$$\mathcal{I}(z_{t+1}^h) = \sum_{m=1}^{M} l_{t+1,m}(z_{t+1}^h)p_{t,m} \log \left( \frac{l_{t+1,m}(z_{t+1}^h)}{l0(z_{t+1}^h)(1 - P1_t) + l1(z_{t+1}^h)P1_t} \right)$$
$$= P1_t l1(z_{t+1}^h) \log \left( \frac{l1(z_{t+1}^h)}{l0(z_{t+1}^h)(1 - P1_t) + l1(z_{t+1}^h)P1_t} \right) +$$

$$(1 - P1_t)l0(z_{+1}^h) \log \left( \frac{l0(z_{t+1}^h)}{l0(z_{t+1}^h)(1 - P1_t) + l1(z_{t+1}^h)P1_t} \right) \tag{11}$$

The mutual information in (10) is now expressed as an integral function of $l0$ and $l1$, and $P1_t$ over $z_{t+1}^h$. If the BCI classifier score pdfs, $l0$ and $l1$, are fixed, after performing the integral over $z_{t+1}^h$, the mutual information is only a function of $P1_t$. *The objective function is now conveniently parameterized by $P1_t$, the total prior probability of characters within a flash group.*

The mutual information function of a BCI user can be estimated from their respective calibrated classifier score pdfs. Figure 2 shows examples of mutual information functions estimated for different BCI users. Given a user's mutual information function, the $P1_t$ value that maximizes the objective function, which we denote as $P1^{\text{opt}}$, can be extrapolated. The advantage of an analytical solution is that, while the flash group that maximizes the objective function varies at every time step given the observed data, the stimulus that maximizes the objective function can simply be defined by its prior probability. *Hence, the flash group whose prior probability mass (9) is closest to $P1^{\text{opt}}$ can be determined and selected for presentation.* With a known solution, we can more easily explore a much larger stimulus space in a time efficient manner by using a greedy search strategy to iterate over an ordered list of character probabilities to construct a close-to-optimal flash group.

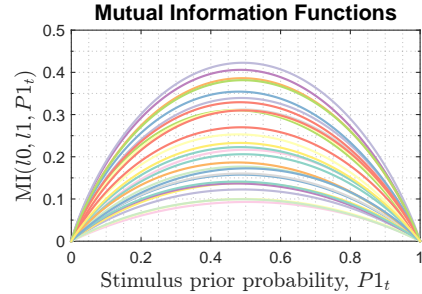

Figure 2: Examples of user-specific mutual information (MI) functions estimated using their respective calibrated BCI non-target $(l0)$ and target $(l1)$ classifier score pdfs $(n = 29)$ [19].

## 2.3 Considerations
## for Real-time Algorithm Implementation

**System Constraints** So far, it is assumed in (6) that the classifier score, $y_t$, that is generated after presenting the flash group, $\boldsymbol{f}_t$, is available instantaneously prior to selecting the flash group for the next time step, $\boldsymbol{f}_{t+1}^s$. However, this is typically not the case during real-time BCI implementation as there is a delay between presenting a stimulus and computing the corresponding classifier score due

to processing a time window of EEG data, as illustrated in Figure 3. Due to the shorter ISI duration relative to the data processing window, stimulus presentation is still ongoing while accumulating the EEG data buffer associated with a current flash group, $\boldsymbol{f}_t$. For example, in Figure 3, the flash groups $[\boldsymbol{f}_{t+1}, ..., \boldsymbol{f}_{t+6}]$ will have already been presented prior to computing $y_t$. We will define the *data processing delay* by the number of additional flash groups prior to being able to compute the classifier score after each flash group presentation, denoted as $\tau$. This means with adaptive stimulus selection, the first $\tau$ flash groups have to be initialized and the data available at time step $t$ is used to select the flash group at $t + \tau + 1$.

To accommodate the data processing delay, the mutual information function is modified accordingly:

$$I(Y_{t+\tau+1}^h; C^* | \boldsymbol{y}_t, \mathbf{f}_{t+\tau+1}^h)$$
$$= \mathbb{E}_{y_{t+\tau+1}^h | \boldsymbol{y}_t, \mathbf{f}_{t+\tau+1}^h} \left[ D_{KL}(p_{t+\tau+1}^h || p_t) \right].$$

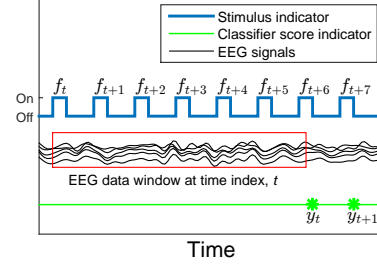

Figure 3: Illustration of the BCI data processing delay between when a stimulus, $f_t$, is presented and when the resulting classifier score, $y_t$, is computed.

However, computing the conditional hypothetical classifier score, $y_{t+\tau+1}^h | \boldsymbol{y}_t, \mathbf{f}_{t+\tau+1}^h$, is computationally expensive: it involves a series of nested integrals over all possible flash groups, $[\boldsymbol{f}_{t+1}^h, ..., \boldsymbol{f}_{t+\tau}^h]$, and classifier scores, $[y_{t+1}^h, ..., y_{t+\tau}^h]$. For computational simplicity, we approximate the data available at $t + \tau$ with that available at $t$, i.e. $\hat{\boldsymbol{y}}_{t+\tau} = \boldsymbol{y}_t$ and $\hat{\boldsymbol{p}}_{t+\tau} = \boldsymbol{p}_t$. While these approximations are not the best estimates for their respective values at $t+\tau$, the substitutions enable the use of (5) when implementing adaptive stimulus selection with a data processing delay:

$$\boldsymbol{f}_{t+\tau+1}^s = \arg \max_{\boldsymbol{f}_{t+\tau+1}^h} \mathbb{E}_{y_{t+\tau+1}^h | \hat{\boldsymbol{y}}_{t+\tau}, \mathbf{f}_{t+\tau+1}^h} \left[ D_{KL}(p_{t+\tau+1}^h || \hat{p}_{t+\tau}) \right], \forall \boldsymbol{f}_{t+\tau+1}^h \in \boldsymbol{\mathcal{F}^c} \qquad (12)$$

**Physiological Constraints** We also consider mitigating the impact of psychophysical factors during stimulus selection, which is achieved by placing restrictions in the stimulus space. Refractory effects are typically mitigated by imposing a minimum time interval between any character's presentation to increase the likelihood of a long TTI to elicit ERPs with high SNRs [19]. Alternatively, refractory effects can be mitigated by using a non-naive classification algorithm that models the relationship between TTI and classifier scores to bias the stimulus selection process towards TTI values that will maximize the discriminability between target and non-target classifier scores. (See supplementary material, Section A for the corresponding objective function.) However, other psychophysical factors, such as adjacency distractions and visual fatigue, are more complex to model for mitigation. Using insight from previous studies, we imposed additional constraints to mitigate the impact of these other effects. For example, the spatial distance between characters in large-sized flash groups usually tends to be small, which may exacerbate adjacency distractions. Also, consecutive presentations of highly correlated flash groups can lead to visual fatigue. We mitigated refractory effects and adjacency distractions by imposing a minimum TTI and limiting the flash group size, respectively.

## 2.4 Pseudo-code for Adaptive Stimulus Selection

Given an initialization flash group set, $[\boldsymbol{s}_1, \boldsymbol{s}_2, ...., \boldsymbol{s}_\tau]$ and classifier score pdfs $l0$ and $l1$, the adaptive stimulus selection algorithm assuming a data processing delay $\tau$ (12) and a search space of flash groups with stimulus space constraints, $\boldsymbol{\mathcal{F}}^c$, is outlined in Algorithm 1.

## 3 Simulation Experiment

**Algorithm 1:** Adaptive BCI Stimulus Selection Using Mutual Information (MI).

**Offline Solution**
$$P1^{\text{opt}} = \arg \max_{P1_t} \text{MI}(l0, l1, P1_t) \qquad \triangleright (10)$$

**Online Encoder**
   **if** $1 \le t \le \tau$ **then**
      $\boldsymbol{f}_t^s = \boldsymbol{s}_t$     $\triangleright$ Initialization set;
   **else**
      $\boldsymbol{f}_t^s = \arg \min_{\boldsymbol{f}_t^h \in \boldsymbol{\mathcal{F}}^c} \left| P1^{\text{opt}} - P1_{t-1-\tau}(\boldsymbol{f}_t^h) \right|$

Using the framework outlined in [23], we performed simulations (in MATLAB) to compare our adaptive stimulus selection algorithm with two

conventional random BCI stimulus selection methods, the RC [4] and the checkerboard (CB) [9] paradigms. The following scenarios with the adaptive stimulus selection algorithm were simulated:

(i) An ideal scenario with a greedy search strategy and no constraints imposed (6) to obtain a performance upper bound.

(ii) A scenario with a greedy search strategy and imposed constraints (12) to obtain more realistic performance bounds associated with real-time algorithm implementation. The constraints included: a data processing delay $\tau = 6$; a flash group size limit of 9; and a stochastic TTI restriction using a pre-defined probability distribution with a minimum value of 3.

(iii) A scenario with a search space of only row and column flash groups and the same constraints as in (ii) above (excluding the flash group limit) to obtain performance bounds when using a more restricted search space. The character selection process occurred in two stages with the selection of the row flash group followed by the selection of the column flash group, or vise versa. We denote this two-stage selection process with row and column flash groups as $RC^2$.

In each simulation run, the target character was uniformly drawn from 72 characters assuming an $8 \times 9$ grid, and flash groups were generated based on the specified stimulus paradigm. The classifier scores associated with non-target and target flash groups were assumed to be normally distributed and drawn with parameters $d = (\mu_1 - \mu_0)/\sigma$, where $d$ is the detectability index, which quantifies classifier performance level; $\mu_0$ and $\mu_1$ are the means of the classifier score pdfs $l0$ and $l1$, respectively, and $\sigma$ is the assumed common standard deviation. In the Bayesian DS algorithm, character probabilities were initialized uniformly, and the stopping threshold and data collection limit were set to $P_{th} = 0.9$ and 120 stimulus flashes, respectively.

Selection accuracy and the average number of flash group presentations prior to character selection, denoted as the expected stopping time (EST), were estimated with results averaged over 1500 simulation runs. Results are shown in Figure 4. (See supplementary material, Section B for example stimulus presentation schedules generated for the stimulus presentation paradigms.) In general, the adaptive stimulus paradigms performed significantly better than the random paradigms, even given the performance drop from the ideal condition. The advantage of a larger search space during adaptive stimulus selection is evident when comparing the performances between the $RC^2$ and greedy adaptive paradigms with constraints. Overall, these results show the significant margins of improvements in BCI accuracy and spelling speed that can potentially be obtained with adaptive stimulus selection.

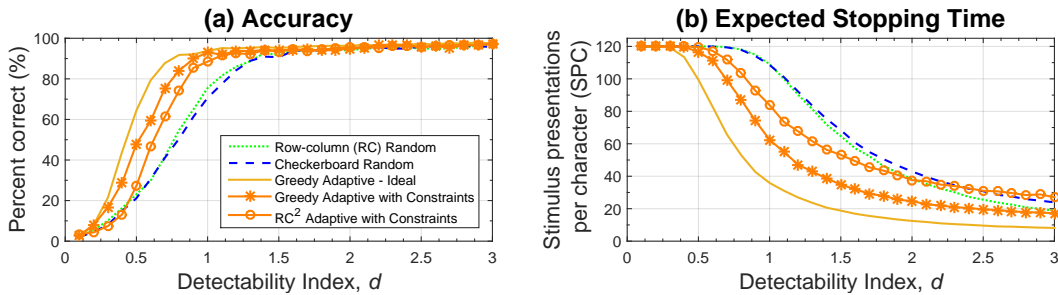

Figure 4: Results from simulations showing the performance of the Bayesian dynamic stopping algorithm with various BCI stimulus presentation paradigms as a function of detectability index, $d$.

## 4 Online Experiment

A preliminary study was conducted to investigate the utility of adaptive stimulus selection during real-time BCI use, as compared to a conventional BCI stimulus selection algorithm. Eight healthy participants were recruited from the student population at Duke University for a study approved by the Institutional Review Board. Participants gave informed consent prior to their experiment session. The open source BCI2000 software [24] was used to implement the P300 speller with the Bayesian DS algorithm, with the stopping probability threshold and data collection limit set to $P_{th} = 0.9$ and 145 stimulus flashes, respectively. The flash duration, ISI and time pause between character selections were set to 62.5 ms, 62.5 ms and 3.5 s, respectively. Data collected from electrodes {Fz, Cz, P3, Pz,

P4, PO7, PO8, Oz} were used for signal processing [25]. EEG signals were sampled at a rate of 256 Hz and filtered between 0.5-30 Hz [19]. Features were extracted from an 800 ms segment of EEG data at each electrode following each flash, by down-sampling to a rate of 20 Hz using bin averaging [25]. The averaged samples were concatenated across channels to form the feature vectors supplied to a classifier. Two stimulus paradigms were tested: the CB paradigm and the greedy adaptive paradigm with constraints (Section 3 (ii)).

Participants performed word copy-spelling tasks (three 6-letter words) with the P300 speller. In a copy-spelling task, the user is instructed by the BCI as to the character in the grid to focus on. The BCI evaluation protocol for a stimulus presentation paradigm included: a calibration block, where the user performs copy-spelling with no classifier use or BCI feedback, to collect EEG data to estimate user-specific BCI parameters (classifier weight vector, classifier likelihood functions and the objective function for adaptive stimulus selection); and a test block, where the user performs copy-spelling with use of the estimated BCI parameters and BCI feedback to evaluate performance. Details on how we calibrated the adaptive stimulus paradigm are provided in the supplemental material, Section C. The testing order of the stimulus paradigms was randomized across participants to avoid order bias.

Participant accuracy and EST results are shown in Figure 5(a) and (b), respectively. Statistical significance tests were not performed due to the small sample size. On average, a small drop in accuracy was experienced with the adaptive paradigm ($M \pm SD = 72.50 \pm 26.11\%$) when compared to the CB paradigm ($77.50 \pm 23.95\%$), with half of the participants maintaining similar accuracy levels in both stimulus presentation paradigms. Across participants, significant reductions in the EST were obtained with the adaptive paradigm ($55.77 \pm 25.70$ stimulus presentations per character (SPC)) when compared to the CB paradigm ($87.12 \pm 46.54$ SPC). User feedback revealed that in the adaptive paradigm, there were persistent presentations of characters in the spatial vicinity of the target character, which was distracting to most users while they focused on the target character.

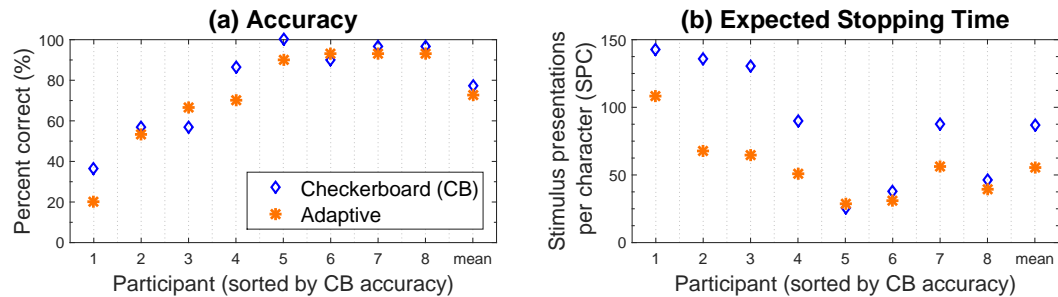

Figure 5: Participant performance, (a) expected stopping time, and (b) accuracy, with the checkerboard and adaptive stimulus paradigms. The mean for each measure is shown on the far right of each panel.

The common user experience of adjacency distractions with the adaptive paradigm condition indicated that the constraints we imposed to minimize the impact of these effects were inadequate and this may have limited the ability to obtain the full performance benefits with the adaptive stimulus paradigm. Based on the stochastic progression of the classifier scores, the mean probability of the target character should increase progressively over time, with a faster rate of convergence in a better-performing stimulus paradigm, assuming the same performance level across paradigms. To inform future algorithm refinements, we performed post-hoc analysis to gain insight into how the relationship between user behavior and stimulus paradigm performance were reflected in the data.

Figure 6 shows the progression of the mean target character probabilities in both paradigms. Half of the participants achieved higher spelling speeds with our adaptive stimulus paradigm while maintaining similar accuracy levels as with the CB paradigm, which is generally illustrated in Figure 6 by the faster algorithm convergence of the mean target character probability in the participants. In the other participants, we observed an initial increase in the rate of convergence with our adaptive algorithm. This indicates the algorithm is still useful in quickly narrowing down the potential choices for the target character and biases the stimulus selection process accordingly by presenting characters in the vicinity of the target character. However, at higher stopping times, the target character probability may occasionally decline and not always recover. We hypothesize that the occasional decline in the mean target character probability at higher stopping times is a negative

consequence of the increased adjacency distractions and user fatigue that occur as a result of the repetitive presentations of characters around the target character in the adaptive stimulus paradigm.

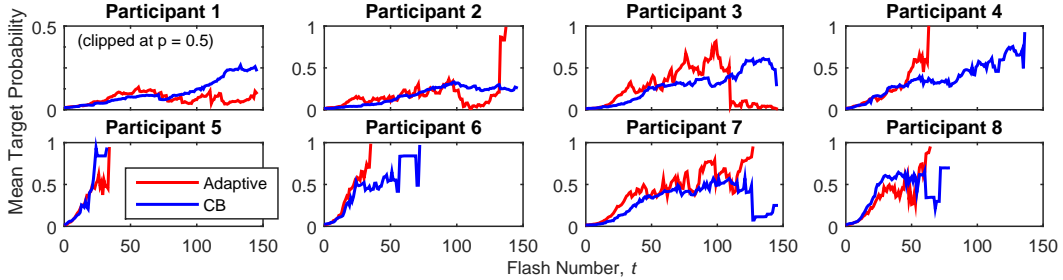

Figure 6: Progression of the mean target character probability for the adaptive and checkerboard (CB) stimulus presentation paradigms, as a function of stimulus flash number for each participant.

We previously highlighted the need to impose constraints in the stimulus space to mitigate the negative impact of psychophysical effects (Section 2.3). The interaction between algorithm and application is crucial for understanding the likelihood of success of proposed methods due to theoretical assumptions that are made during algorithm development. Adjacency distractions are more complex to model and mitigate, and the constraint we imposed of limiting flash group size was not enough to overcome the adjacency issue. Better approaches to mitigate adjacency distractions include imposing spatial constraints in the stimulus space based on the interface geometry to prevent characters that are directly adjacent to each other from flashing together, or using a center interface layout where singletons or groups of characters are sequentially presented in the center of a screen [26].

# 5 Conclusions

We have developed and tested an adaptive stimulus selection algorithm for the P300 BCI speller that utilizes previous user responses to select future stimuli that are maximally informative of the user's intent to improve BCI communication efficiency. To our knowledge, this is the first adaptive BCI stimulus selection method with a tractable analytical solution that provides the flexibility to efficiently sample the high dimensional stimulus space in a real-time feasible manner. We provide a simple parameterization of our objective function in a one-dimensional space that quantifies the prior probability mass of a stimulus under consideration, irrespective of its content. We outlined practical steps to account for BCI system computational limitations and the potential impact of psychophysical factors during stimulus selection with real-time algorithm implementation. Although this work focuses on the P300 speller, the methodology developed here for adaptive stimulus selection is applicable to other BCIs that execute a number of possible action queries prior to decision-making.

While the performance trends in the simulations were not entirely reflected with measures of online BCI use, our preliminary findings are promising as they show the potential benefit of using a data-driven stimulus selection strategy. In general, faster convergence was achieved with the adaptive stimulus selection algorithm; from the post-hoc analysis, we hypothesize that the initial increase of the target character probability indicates that the stimulus selection process was able to quickly narrow down to the vicinity of the target character. We believe that the limitations associated with a grid interface layout can be addressed with additional refinements of the stimulus space constraints, such as imposing spatial restrictions on characters within a flash group to minimize adjacency distractions. The ability to search a larger stimulus space with our algorithm allows for greater flexibility to achieve the best possible implementation of an adaptive stimulus paradigm with a given BCI configuration.

It should be noted that while BCI studies are typically conducted in a non-disabled population for time efficiency and practicality during algorithm development, results from a non-disabled population may not necessarily be applicable to individuals with severe neuromuscular limitations due to differences in disease cause and level of neuromuscular control. We expect different tolerance levels to stimulus presentation properties in a disabled population where there is a more limited ability to easily navigate the user interface. Future work includes additional refinements and testing of the adaptive stimulus selection algorithm with a larger sample size of non-disabled participants, and validating the algorithm in a study with individuals with ALS.

**Author Contributions**    DK conceptualized and developed the adaptive stimulus selection algorithm, designed and conducted the simulation and the real-time human BCI experiments. BOM, LMC and CST provided technical contributions to the theoretical development and implementation of this work. SL performed the post-hoc analysis. All authors contributed to writing and editing the manuscript.

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
