[Supplementary Material]

# Supplementary Material for Information-based Adaptive Stimulus Selection to Optimize Communication Efficiency in Brain-Computer Interfaces

**Boyla O. Mainsah,**[1] **Dmitry Kalika,**[2] **Leslie M. Collins,**[1,*]
**Siyuan Liu,**[1] **Chandra S. Throckmorton**[1]
[1]Department of Electrical and Computer Engineering, Duke University, Durham, NC, USA
[2]Johns Hopkins University Applied Physics Laboratory, Laurel, MD, USA
[*]Corresponding author: leslie.collins@duke.edu

## A    Adaptive Stimulus Selection with a Non-naive Classification Algorithm

We consider the non-naive Bayesian algorithm for the P300 BCI speller where the target stimulus event classifier scores are dependent on the time interval between target character presentations. Let [...TNN**T**...] represent a character's presentation pattern with a target-to-target interval (TTI) value of 3, where N denotes a non-target stimulus event; T denotes a target stimulus event; and **T** denotes the target stimulus event under consideration. The character probabilities are updated accordingly:

$$P(C^* = m|\boldsymbol{y}_t, \mathbf{f}_t) = \frac{p(y_t|C^* = m, \mathbf{f}_t)P(C^* = m|\boldsymbol{y}_{t-1}, \mathbf{f}_{t-1})}{\sum_j p(y_t|C^* = j, \mathbf{f}_t)P(C^* = j|\boldsymbol{y}_{t-1}, \mathbf{f}_{t-1})}, \tag{A.1}$$

$$p(y_t|C^* = m, \mathbf{f}_t) = \begin{cases} l0(y_t), & \text{if } f_{t,m} = 0 \\ l1(y_t, i_{t,m}), & \text{if } f_{t,m} = 1 \end{cases}, \tag{A.2}$$

where $P(C^* = m|\boldsymbol{y}_t, \mathbf{f}_t)$ is the posterior probability that character $m$ is the target character, $C^*$, given the presented flash groups, $\mathbf{f}_t = [\boldsymbol{f}_1, \ldots, \boldsymbol{f}_t]$, and classifier scores, $\boldsymbol{y}_t = [y_1, \ldots, y_t]$ at time index $t$; $P(C^* = m|\boldsymbol{y}_{t-1}, \mathbf{f}_{t-1})$ is the prior probability; $p(y_t|C^* = m, \mathbf{f}_t)$ is the likelihood of generating the classifier score, $y_t$, given character $m$ is the target character and the current flash group sequence, $\mathbf{f}_t$; $l0$ is the classifier score pdf for non-target stimulus events; $i_{t,m}$ is TTI of character $m$ assuming that $m$ is the target character at time index $t$; and $l1(y_t, i_{t,m})$ is the likelihood density of a classifier score, conditioned on $i_{t,m}$, if $m$ is presented at $t$.

Similar to (9), for a flash group at time index $t + 1$, the character prior probabilities at time index $t$ can be grouped based on their respective TTI values:

$$P1_{t,\text{TTI}}(\boldsymbol{f}_{t+1}^h) = \sum_{\forall c: \boldsymbol{f}_{t+1,c}^h = 1 \,\cap\, i_{t+1,c} = \text{TTI}} p_{t,c} \tag{A.3}$$

$$\boldsymbol{P1}_t(\boldsymbol{f}_{t+1}^h) = \left[ P1_{t,1}(\boldsymbol{f}_{t+1}^h), P1_{t,2}(\boldsymbol{f}_{t+1}^h), ..., P1_{t,K}(\boldsymbol{f}_{t+1}^h) \right] \tag{A.4}$$

where $P1_{t,\text{TTI}}(\boldsymbol{f}_{t+1}^h)$ is the sum of prior probabilities at time index $t$ for the characters that are flashed in $\boldsymbol{f}_{t+1}^h$ with the given TTI value; $\boldsymbol{P1}_t(\boldsymbol{f}_{t+1}^h)$ is a *vector of the prior probability masses* of characters in the flash group $\boldsymbol{f}_{t+1}^h$ grouped by their respective TTI values, $[1, ..., K]$; and $K$ is the maximum TTI value in the classification model.

Let $\boldsymbol{P1}_t \triangleq \boldsymbol{P1}_t(\boldsymbol{f}_{t+1})$ for notational simplicity. The corresponding mutual information (MI) function parameterized by the probability vector $\boldsymbol{P1}_t$ is:

$$I(Y_{t+1}^h; C^* | \boldsymbol{y}_t, \mathbf{f}_{t+1}^h) = \int_{-\infty}^{\infty} \mathcal{I}(z_{t+1}^h) dz_{t+1}^h \tag{A.5}$$

$$\mathcal{I}(z_{t+1}^h)$$

$$= \sum_{\forall \text{TTI}} P1_{t,\text{TTI}} l1(z_{t+1}^h, \text{TTI}) \log\left(\frac{l1(z_{t+1}^h, \text{TTI})}{l0(z_{t+1}^h)(1 - \sum_{\forall \text{TTI}} P1_{t,\text{TTI}}) + \sum_{\forall \text{TTI}} P1_{t,\text{TTI}} l1(z_{t+1}^h, \text{TTI})}\right) +$$

$$\left(1 - \sum_{\forall \text{TTI}} P1_{t,\text{TTI}}\right) l0(z_{+1}^h) \log\left(\frac{l0(z_{t+1}^h)}{l0(z_{t+1}^h)(1 - \sum_{\forall \text{TTI}} P1_{t,\text{TTI}}) + \sum_{\forall \text{TTI}} P1_{t,\text{TTI}} l1(z_{t+1}^h, \text{TTI})}\right).$$

$$\tag{A.6}$$

**Pseudo-code for Adaptive Stimulus Selection**   Similar to Algorithm 1, we can determine the flash group whose corresponding mutual information value is closest to that which maximizes the mutual information (MI) function (A.5). Given an initialization flash group set, $[\boldsymbol{s}_1, \boldsymbol{s}_2, ...., \boldsymbol{s}_\tau]$ and classifier pdfs $l0$ and $\{l1_{\text{TTI}}\}_{\text{TTI}=1}^K$, the adaptive stimulus selection algorithm assuming a data processing delay $\tau$ and a search space of flash groups with stimulus space constraints, $\mathcal{F}^c$, is outlined in Algorithm A.1.

---

**Algorithm A.1:** Adaptive BCI Stimulus Selection Using Mutual Information (MI) Assuming a Non-naive Bayesian Algorithm

---

**Offline Solution**
$$\text{EIG}_{\max} = \max_{\boldsymbol{P1}_t} \text{EIG}\left(l0, \{l1_{\text{TTI}}\}_{\text{TTI}=1}^K, \boldsymbol{P1}_t\right)$$
**Online Encoder**
    **if** $1 \leq t \leq \tau$ **then**
        $\boldsymbol{f}_t^s = \boldsymbol{s}_t$                       ▷ Initialization set;
    **else**
        $\boldsymbol{f}_t^s = \arg\min_{\boldsymbol{f}_t^h \in \mathcal{F}^c} \left| \text{EIG}_{\max} - \text{EIG}\left(l0, \{l1_{\text{TTI}}\}_{\text{TTI}=1}^K, \boldsymbol{P1}_{t-1-\tau}(\boldsymbol{f}_t^h)\right) \right|$

---

By explicitly accounting for TTI-related effects in the classification model, the BCI stimulus selection process is inherently biased to create flash groups with characters whose TTI values are likely to generate ERPs with high signal-to-noise ratios while also minimizing spelling spelling speed. While TTI values can take any positive integer value, it is recommended that a small number of values be considered for the classification model to generate enough training samples during system calibration and obtain good estimates for the TTI-specific classifier likelihood pdf [1].

Assuming TTI values $[1, 2, 3]$ for simplicity, three cases of varying discriminability between the TTI-specific classifier scores and the non-target classifier scores are presented in Figure A.1 to illustrate how stimulus selection can be affected by incorporating TTI in the MI function. In this case, the MI (volume) function is parameterized by the TTI-specific prior probability vector $\boldsymbol{P1}_t = [P1_{t,1}, P1_{t,2}, P1_{t,3}]$. In the top row in Figure A.1 where the TTI-specific likelihood pdfs are identical, the MI function is maximized when $P1_{t,1} + P1_{t,2} + P1_{t,3} = 0.5$. Note that in this scenario, the stimulus selection process is similar to the case where there is only one target likelihood pdf, which is biased to a large extent towards flash groups with characters with a TTI value of 1. In the middle row, the TTI-specific pdfs for the TTI values 2 and 3 are identical and have higher discriminability from the non-target likelihood pdf compared to the TTI-specific pdf for a TTI value of 1. The MI function is maximized when $P1_{t,2} + P1_{t,3} = 0.5$ and $P1_{t,1} = 0$. In this scenario, the stimulus selection process will be biased to a large extent towards flash groups with characters with a TTI value of 2. In the bottom row, the TTI-specific pdfs are distinct from each other, with increasing discriminability from the non-target pdf as TTI increases. The MI function is maximized when $P1_{t,3} = 0.5$ and $P1_{t,1} = P1_{t,2} = 0$. In this scenario, the stimulus selection process will be biased towards flash groups whose characters have a TTI value of 3.

Figure A.1: Examples of classifier score likelihood probability density functions (pdfs) for non-target and target scores with target-to-target intervals (TTIs) 1, 2, and 3 and their corresponding mutual information volumes, $\text{MI}(l0, \boldsymbol{l1}_{\text{TTI}}, \boldsymbol{P1}_t)$. Each row shows the likelihood pdfs on the left and the mutual information volume for the corresponding likelihood pdfs on the right. In the top row, the target likelihood pdfs for all TTI-bins are equivalent. In the middle row, the TTI-specific pdfs for TTI 2 and 3 are the same. In bottom row, all three TTI values have different pdfs.

# B    Stimulus Presentation Schedules

Simulations of P300 spelling runs were performed to initially assess the potential utility of the adaptive stimulus algorithm in improving BCI performance (see Section 3 for details). The stimulus presentation paradigms were implemented assuming a 72-character user interface with an $8 \times 9$ grid layout, such as the example shown in Figure B.1. Example stimulus presentation schedules generated for the different stimulus presentation paradigms are shown in Figure B.2.

Figure B.1: Example P300 BCI speller interface.

Figure B.2: Example stimulus presentation schedules for different stimulus paradigms generated during P300 speller simulations assuming a $8 \times 9$ grid interface, such as the example shown in Figure B.1. The characters are numbered accordingly: characters A to H are numbered $1 - 8$; characters I to P are numbered $9 - 16$, etc. Each column represents a flash group, with the characters presented highlighted. Each row represents the stimulus presentation pattern or *codeword* of a character.

## C   Online Calibration of the Adaptive Stimulus Selection Paradigm

As with any supervised learning algorithm, it is crucial that the training data is generated from a similar underlying distribution as the testing data. To ensure that the EEG data between training and testing tasks are similar, the experimental paradigms used for stimulus presentation in both tasks must also be similar. During the training task for a random stimulus presentation paradigm, the flash groups are randomly selected and presented to the user as they would be during the testing task, as is the case with the checkerboard paradigm [2]. However, the training task for the adaptive stimulus paradigm is less straightforward as the stimulus selection algorithm chooses stimuli to present based on character probabilities obtained by using calibrated BCI system parameters (classifier, classifier likelihood pdfs and mutual information function). However, during the training phase, the user-specific classifier models are not yet calibrated.

To enable EEG data collection during the training phase without BCI system calibration, synthetic data were used to guide the stimulus selection process during the training task. Conditioned on the target character during copy-spelling, non-target and target classifier scores associated with flash group presentations were randomly generated from pre-defined pdfs and the greedy adaptive algorithm was then used to select stimuli to present based on the drawn classifier scores and pre-defined data likelihood pdfs. Assuming normality, let $\{\mu_0, \sigma^2\}$ and $\{\mu_1, \sigma^2\}$ represent the mean and variance parameters of the pre-defined non-target and target classifier score pdfs, respectively. Based on simulations, a classifier detectability index [3] of $d = \frac{\mu_1 - \mu_0}{\sigma^2} = 1$ was chosen (with parameters $\{\mu_1 = 1, \mu_1 = 0, \sigma^2 = 1\}$) as a suitable performance level that results in enough target and non-target stimulus event instances to extract labeled features for classifier training.