[Reviews · NeurIPS 2018]

Reviewer 1



The manuscript presents an adaptive stimulus selection strategy for P300 BCI. The novelty of the approach is the tractable analytical solution that allows sampling the stimulus space during online BCI use. Preliminary results with 8 study participants suggest the feasibility of the proposed approach. The strength of the manuscript is idea to use data driven approaches for stimulus presentation and that an analytical solution can be computed online. The method is original and clearly outlined. The weakness is that with decreasing stimulation time, also the performance decreases. This limits the quality and usefulness of the proposed method. This also raises the question on which method allows more reliable communication. Faster communication with higher errors or slower but more reliable classification. Would be interesting to learn about this relationship. This would allow reader to put the results in proper perspective and may also allow to judge the validity of the authors claims. Response to the author's rebuttal: I thank the authors for their detailed response. The reviewer, However, still remains comfortable with the initial overall score.

Reviewer 2



The submission could have been written with more clarity despite an interesting proposed framework. At times, the language and phrasing comes across more convoluted than how it could have been expressed. However, the proposed method is creative in its approach and would be a valuable contribution to the BCI field. Addressing constraints and concerns with the method proposed would also have been welcome, alongside the proposed future research within ALS.

Reviewer 3



The authors consider the problem of adaptively selecting stimulus presentations online for a BCI based on evoked potentials. They focus on the problem of presenting the best “flash groups” in a P300 spelling application. They derive a method to select stimuli that maximize mutual information between a target character and the information likely to be extracted from EEG signals (in the form of classifier scores). They derive their methods with an eye towards computational efficiency to allow their method to work online during BCI use and demonstrate their method with simulated and real, online experiments. The authors do a very nice job of motivating the need for online stimulus selection for evoked BCI work and it is clear the authors are addressing an important problem. However, I feel the paper, in it’s current form, may not be appropriate for NIPS. My first concern is the work presented here may be very application specific. It’s not clear how the methods presented here would generalize to problems outside of the evoked BCI paradigm. Additionally, it seems that many of the technical contributions of the paper (in particular the form of optimization objective) are motivated by the need to run their algorithm in real time and leverage special structure (binary representation) specific to the P300 spelling paradigm. My other second concern with the paper is that while the authors have simplified the objective function, it’s not clear to me the importance of this. The search space is still (if I understand correctly) over a set of 2^M possible stimulus configurations, which can potentially be quite large. It seems the authors are currently searching this space with a Greedy algorithm. It then seems the authors have turned a hard search problem into a slightly less difficult one, but from a technical point of view, it seems this may be an incremental improvement. Finally, the authors are to be commended for honestly reporting their results on real data, but as they point out, the current results leave room for improvement, as classification accuracy of the target stimuli with their adaptive strategy decreases with time. The authors indicate they have good reason to believe they may understand the cause of this, but it would be helpful to see this demonstrated in practice. Finally, a minor concern with the paper is one of technical clarity. I found the derivations throughout sections 2.2 – 2.3 a bit hard to follow. For example, it is not clear to me that 3a is the correct equation for the conditional mutual information between Y_{t+1}^h and C*. It may very well be, but it seems to be there is a term missing in the denominator of the log term. Response to authors rebuttal: I thank the authors for their detailed and clear response. After reviewing their response, I now see how their formulation of the objective fundamentally reduces the complexity of the objective which must be calculated during run-time. Additionally, while I still find the performance during real experiments somewhat concerning, I also acknowledge the substantial decreases in expected stopping times achieved with their method in these real experiments. For this reason, I am revising my overall score upward.